# Study on awareness and management based health action using video intervention (SAMBHAV) for postpartum depression among mothers attending immunisation clinic in a tertiary medical college hospital: Study protocol

Latha K.[1]ᴼ*, Sundarnag Ganjekar[2]ᴼ, Meena K. S.[1], Virupaksha H. S.[3], Mariamma Philip[4]‡, Suman G.[5], Dinesh Rajaram[5], Swathi Acharya[1]‡, Kimneihat Vaiphei[6], Somshekhar A. R.[7]

1 Department of Mental Health Education, National Institute of Mental Health and Neuro Sciences, Bengaluru, Karnataka, India, 2 Department of Psychiatry, National Institute of Mental Health and Neuro Sciences, Bengaluru, Karnataka, India, 3 Department of Psychiatry, Ramaiah Medical College, Bengaluru, Karnataka, India, 4 Department of Biostatistics, National Institute of Mental Health and Neuro Sciences, Bengaluru, Karnataka, India, 5 Department of Community Medicine, Ramaiah Medical College, Bengaluru, Karnataka, India, 6 Department of Psychiatric Social Work, National Institute of Mental Health and Neuro Sciences, Bengaluru, Karnataka, India, 7 Department of Pediatrics, Ramaiah Medical College, Bengaluru, Karnataka, India

ᴼ These authors contributed equally to this work.
‡ MP and SA also contributed equally to this work.
* latha12k@gmail.com

## Abstract

### Introduction

Pregnancy exerts a detrimental effect on women's mental health. Maternal mental health is considered as one of the public health concerns as it impacts the health of both mother and the child. One in five people in developing countries experience serious mental health issues during pregnancy and after giving birth. In India, postpartum depression (PPD) affects 22% of women, according to a research by WHO. The available data on mental health literacy among women, showed that only 50.7% of the postpartum mothers who were attending paediatric tertiary care centres had adequate knowledge about PPD. It is crucial to diagnose early and adequately manage postpartum depression to avoid long-term consequences. It is also essential to seek help and utilise the available resources and services to avoid worsening of the condition and to aid in the recovery. This demonstrates the need to promote awareness, improve help seeking, reduce stigma and treatment gap associated with PPD through educational video intervention specific to cultural context and beliefs.

### Materials and methods

This is a quasi-experimental study without a control group that attempts to improve the awareness among the mothers about postpartum depression to understand better about

**Data Availability Statement:** No datasets were generated or analyzed during the current study. All relevant data from this study will be made available upon study completion.

**Funding:** The proposal has been funded by ICMR as an Ad-hoc project under extramural research program for the year 2021 during the cycle 18.11.2021 to 17.12.2021. (Reference No: 2021-13266).

**Competing interests:** The authors have declared that no competing interests exist.

**Abbreviations:** PHQ 9, Patient Health Questionnaire 9; PoDLiS, Postpartum Depression Literacy Scale; PPD, Postpartum Depression.

the condition and also its management through video intervention. The video intervention will be developed in regional language specific to the cultural context of the setting. The video script will be finalised from the findings of the available literature and also through focus group discussion among mothers and health care professionals which will be analysed qualitatively using thematic identification. The study will use a standardized Postpartum Depression Literacy Scale (PoDLIS) which will be quantitatively analysed using paired t test before and after the intervention. Repeated measures of ANOVA will also be used to analyse the changes in literacy scale scores with respect to socio demographic variables. The mothers will also be screened for PPD using Patient Health Questionnaire 9 (PHQ 9) and feedback will be collected and analysed to find the overall usefulness of video.

## Discussion

If it becomes apparent that this video intervention is successful in raising awareness of PPD among postpartum mothers and reducing stigma, it can be used to aid early identification of mothers with PPD which can result in early management and improved health outcome for both mothers and children. The major goals of the video intervention are to raise awareness, lessen stigma, and prevent PPD through strong family support, adopting healthy lifestyles, having access to information, practising self-care, and enhancing help-seeking.

## Trial registration

The trial is registered under the Clinical Trial Registry- India (CTRI) (CTRI/2023/03/050836). The current study adheres to the SPIRIT Guidelines [See S1 Checklist: SPIRIT Guidelines].

## Introduction

During pregnancy, a woman undergoes several physical, emotional, hormonal, and psychological changes [1]. The mother's world both familial and interpersonal undergoes enormous transformation. A mother may feel a range of emotions after giving birth, from excitement and pleasure to grief and tears [2]. Pregnancy is the time when a female is most vulnerable to developing mental illness. Maternal Mental Health disorders includes; depression, anxiety and psychosis which can develop during pregnancy or after delivery. These illnesses can have detrimental effects on the mother, her child, her family, and society if not treated [3]. Globally around 13% of postpartum mothers and 10% of pregnant women experience a mental disorder, primarily depression [4]. According to a WHO report the prevalence of Postpartum Depression is 22% in India [5]. Only about 50.7% of the postpartum mothers attending a paediatric tertiary care centre in India had adequate knowledge on postpartum depression indicating low mental health literacy [6]. Postpartum Depression can start soon after childbirth or as a continuation of antenatal depression and needs to be treated. It is characterized by symptoms of consistent low mood, inability to bond with the baby, excessive crying, inability to sleep (insomnia) or sleeping too much, overwhelming fatigue or loss of energy, reduced interest and pleasure in activities mother used to enjoy, intense irritability and anger, fear of not being a good mother, hopelessness and thoughts of harming oneself and the baby [7, 8]. A systematic review by Daria et al. showed that the studies on knowledge about PPD showed that around 30.2–62.2% among the public could correctly identify the typical symptoms of PPD and percentage of women reporting difficulties in mother child relationship also varied widely

between 5 and 77.1% [9]. Postpartum Depression places mothers at increased risk for social isolation due to a lack of energy, fatigue, and feelings of incompetence, worthlessness, and helplessness [10]. Major risk factors for Postpartum Depression include past history of psychiatric illness in mothers, financial difficulties, marital conflict, lack of support from husband, domestic violence (DV), intimate partner violence (IPV), stressful life events and poor social support [11]. The birth of a female child can also be considered a risk factor for PPD in India [12, 13], China [14] and Egypt [15].

Postpartum Depression if untreated can predispose to chronic or recurrent depression, this may affect the mother and child relationship, growth and development of the baby [16, 17]. Studies have showed that infants of depressed mothers have less expressive language and performed poorly on measures of cognitive linguistic functioning [18]. Children of mothers with postpartum depression have greater cognitive, behavioural and interpersonal problems compared with the children of non-depressed mothers [19]. Scarcity of available mental health resources, inequities in their distribution and inefficiencies in their utilization are key obstacles to optimal mental health services, especially in low resource countries [20]. The systematic review by Daria et al. mentioned that the structural, attitudinal, and knowledge-related barriers were reported for not seeking help among perinatal women. The structural barriers included cost of treatment and inability to attend appointments due to: time constraints, logistics/transportation, childcare, distance/geographic mismatch, and unavailability of providers/resources. The most common attitudinal barriers were found to be associated with stigma and shame [9, 21, 22].

A research study by Heh et al. looked at the impact of informational assistance in lowering Edinburgh Postnatal Depression Scale scores and the findings showed that providing women with adequate information on postpartum depression during the postpartum period may benefit their psychological well-being [23].

Postpartum Depression often goes undetected as there is lack of awareness regarding maternal mental health issues especially in a country like India. It is imperative to raise public awareness about postpartum depression due to its detrimental effects on both mother and child's overall health [24]. It is also equally important to make timely diagnosis and management of Postpartum Depression to prevent long term consequences. Although pregnancy and childbirth may be the same everywhere, postpartum may be viewed and experienced rather differently by women from different cultures [25]. When considering the Indian context there is a lot of stigma associated with mental illness especially when it comes to maternal mental health. Women frequently express discomfort about getting professional help and worry about being labelled "bad mothers" if they admit having depression [26]. A study by Branquinho et al. suggested that 11.4% of the participants agreed that 'women with postpartum depression cannot be good mothers' and 12.1% agreed with 'women have postpartum depression because they have unrealistic expectations about caring for a baby' [27].

It is crucial to seek proper assistance from the available resources for preventing the PPD from worsening and also to aid in recovery. This highlights the need for initiatives to reduce stigma and encourage help-seeking behaviour [28].

A systematic review of research on video-based educational interventions for modifying health behaviours demonstrated their effectiveness across a variety of health-related topics [29]. A recent study by Chaudary L et al. indicates that the knowledge on post-natal care among pregnant women visiting district hospital for a prenatal check-up can be improved through customised social media-based health education guided by a health belief model [30]. Public educational campaigns highlighting the significance of perinatal mental health problems could counteract misconceptions and trivializing notions as suggested by Daria et al. [9].

Since there is poor awareness, wide treatment gap in lower and middle income countries and lack of interventions, the present study attempts to develop and assess a video-based educational intervention that promotes awareness on Postpartum Depression among mothers in the postpartum period. Studies have showed that the employment of video-based educational interventions improved postpartum depression symptom detection, decreased stigmatizing views, and raised knowledge of the dangers that could affect the mother and baby [31]. In addition, studies on postpartum depression and video-based therapies have also been applied to enhance mother-child interactions [32].

There are no studies available in India which focuses on development of a video based intervention for promoting awareness on PPD among mothers. And the video interventions available from other countries may not be suitable to our setting because of the diverse cultural context, risk factors, language barriers and beliefs associated with PPD in India. Hence, this study is being planned to develop and evaluate a culturally relevant video intervention to improve awareness and reduce stigma on Postpartum Depression among postpartum mothers. The developed educational video series could be utilized for mothers with focus on early identification, self-care and better management of PPD.

## Objectives

1. Development and evaluation of a video based educational intervention in improving the awareness and reducing stigma about Postpartum Depression among postpartum mothers.

## Methods and analysis

### Study design

This study follows Quasi- experiment single group without control design with video intervention to study the effect of video intervention in improving awareness and reduction of stigma on Postpartum Depression among postpartum mothers.

### Study participants

Postpartum mothers within 6 months after child birth attending immunization clinics in urban and rural field practice area of Ramaiah Medical College, Bangalore.

### Inclusion criteria

1. Postpartum mothers within 6 months after childbirth who are willing to give consent and able to understand and speak in Kannada or English.

### Exclusion criteria

1. Mothers diagnosed with any of the major illness or Mental Health conditions.

**Study setting.** Immunization Clinics in Urban and Rural field practice area of Ramaiah Medical College Hospital, Bangalore. As the immunisation clinics are ideal for undertaking any intervention studies among mothers, as they will be visiting these clinics for vaccination of the child.

**Study duration.** 27 Months (December 2022 to February 2025).

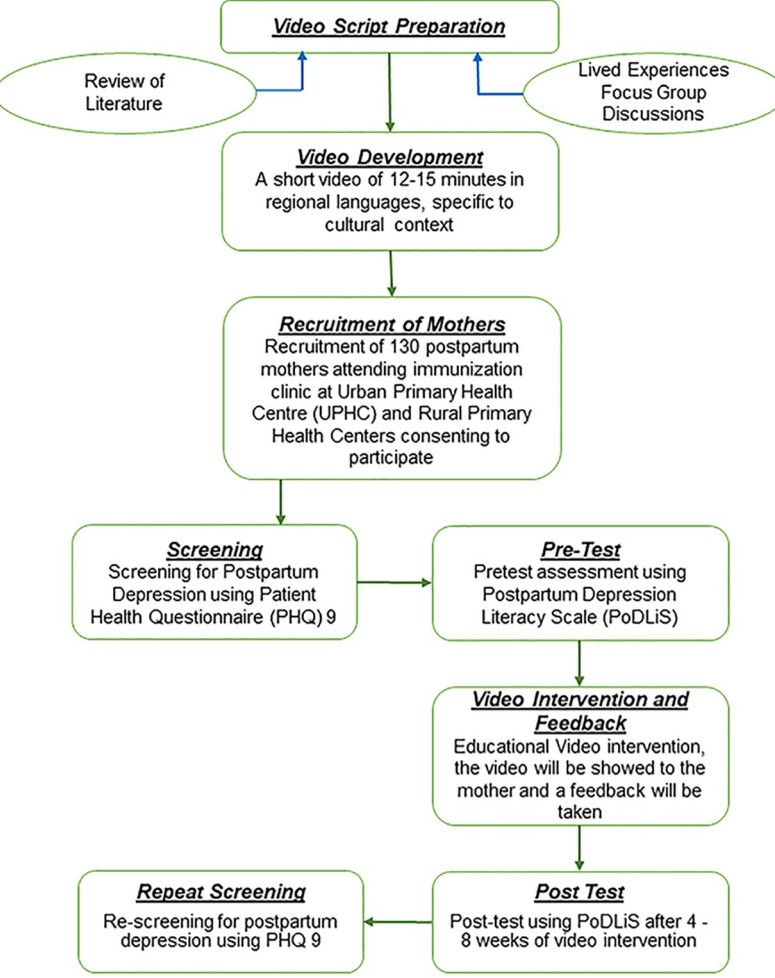

**Fig 1. Flow chart depicting the sequence of project implementation plan.**

**Study sampling.** The postpartum mothers who are attending the immunization clinic at urban and rural health centres under the medical college hospital will be selected using convenience sampling.

**Sample size.** The sample size was calculated based on the awareness of PPD referring to the study by Lekie Dwanyen, Postpartum Depression: Novel use of Video-based interventions using G-power software version 3.1, the effect size was 0.25, at confidence level of 95% and power is 80% for which the sample size is 128 [31]. Here, in this study we will be considering 130 mothers who will be satisfying the inclusion criteria and willing to consent (Fig 1).

## Project implementation plan

1. **Socio-demographic data sheet:**
   Socio-demographic data sheet for mothers visiting the immunization clinic will be developed. This will include the respondents' identification data, viz., age, education, and place of residence, marital status, family type (joint, nuclear, extended) and child birth details.

2. **Postpartum Depression Literacy Scale (PoDLiS)**

The PoDLiS will be used to assess the effectiveness of the video intervention on PPD. PoDLiS is a standardised and validated tool developed by Mirsalimi et al. in 2000 with a Cronbach alpha coefficient of 0.78 [33]. The formal permission has been obtained from the authors for utilising the PoDLiS for this study. It is a 31-item scale that uses 27 variables to measure literacy on the following 7 constructs: the ability to identify postpartum depression, knowledge of risk factors and causes, knowledge and belief of self-care activities, knowledge about professional help available, beliefs about professional help available, attitudes that foster recognition of postpartum depression and appropriate help-seeking, and knowledge of how to look up information related to postpartum depression. Each item is scored on a Likert scale of 1 to 5, with 1 denoting strongly disagree and 5 strongly agree, is used to grade each item. Reverse scoring items receive the opposite score.

The PoDLiS has been used by Poreddi et al. in India to estimate the PPD literacy among postpartum mothers [6]. The tool has not been validated so far for use in India. As a part of the current study the PoDLiS will be translated to Kannada (Vernacular language) and the validity and reliability testing of the same will be undertaken before using the tool for data collection.

3. **Patient Health Questionnaire 9 (PHQ 9)**

The study focuses on postpartum mothers within six months of child birth and who are at risk for PPD. The PHQ 9 tool will be administered to screen the mothers for depression in the study. The PHQ 9 was developed and validated by Kroenke K et al. the internal consistency Cronbach's alpha coefficient of the tool is 0.86. The mothers will be screened using PHQ 9 in the beginning of the trial and also after 4 to 8 weeks during the post test, as there is a risk that some of them may have later developed the symptoms. This depression assessment instrument is self-administered. It combines other prominent major depressive symptoms with DSM-IV depression criteria in a quick self-report tool that is frequently used for screening and diagnosis [34]. There are nine short statements that make up the scale. A mother selects one of the four options that best describes her feelings during the previous two weeks. Mothers who score 5 or more of the nine depressive symptoms (score >15) are diagnosed with major depression, which requires medical intervention.

4. **Feedback Form**

After watching the video, the mothers will receive a feedback form to rate the videos' overall presentation and usefulness including the content, story line, language, audio and video quality, message etc. using a 5 point Likert scale.

## Procedure

1. **Development of Video Intervention**

The steps involved in video development will be as follows:

**(i) Development of script**

a) Literature review: The script for the video will be based on a thorough literature search using PubMed, Psycinfo, Elsevier, Google scholar. The key words for the literature search will be Postpartum Depression, experiences, risk factors, identification, stigma, social support and management. Articles published over the past 10 years in peer reviewed indexed publications in the English language or providing an abstract (with complete information) will be included. Also the articles and blogs published in English/Kannada/Hindi on the

website of the Perinatal Psychiatry services in India focussing on factors associated with Postpartum Depression will be considered. All the articles and blogs considered will be screened for authenticity of information by the perinatal psychiatry experts in the team.

b) Focus Group Discussion (FGD): Separate FGD's among the mother's and the healthcare experts will be conducted. The minimum of 5 to maximum 10 members will be considered for the FGD's. The FGD's are planned to be conducted in 2 rounds or until the intended saturation level has been achieved. The mothers within 18 months of child birth will be considered. The healthcare experts include both doctors and non-doctors working in the field of Perinatal mental health such as Psychiatry, Psychiatric Social Work, Clinical Psychology, Nursing, Obstetrics, Pediatrics, Public Health, Yoga etc. The FGD among mothers will be conducted in the immunization clinics at urban and rural Primary health centers. The FGD among health care experts will be conducted online using video conference platform for the ensuring participation of multidisciplinary team located in different geographical distribution. Separate FGD guides will be developed covering topics on prevalence, risk factors, presentation, barriers and support, and role of educational interventions in prevention and management of PPD. The same will also be qualitatively analyzed.

### (ii) Finalising the script

The final script will be in the form of a story line about the experience of a mother with PPD covering the aspects of the symptoms, stigma associated, barriers for help seeking and management. The script will be finalised based on the literature review and the findings from focus group discussions with mothers and the experts, so that the relevant cultural context is being considered. The developed script will be finalised in discussion with the experts in the internal project team.

### (iii) Writing the 'shooting script'

The finalized script will be transformed into a shooting script of around 10–12 pages with inputs from the video Production team to get a final video of 12–15 minutes' duration. The shooting script will include aspects of filming such as camera angles and cut or fade instructions from one actor to another or to experts.

### (iv) Video production

The characters for the videos as well as the locations for the shoots will be identified. Written informed consent will be obtained from the experts and actors. The video will be made as such that it is relatable for those receiving the intervention. The characters chosen for the study will be sensitive and act in a manner that does not hurt the feelings of anyone. The videos will be recorded and edited as per the requirement.

### (v) Finalising the video

The video will be finalised by the internal team of experts and later will be screened among experts working in perinatal mental health and mothers and the valid suggestions will be incorporated.

## Data collection

The Postpartum Depression literacy scale will be used to conduct the pre and post-test among the postpartum mothers to examine the effectiveness of the video intervention in terms of improving awareness and reducing the stigma. The literacy scale is a self-report measure which will be easy to administer, however in case of mothers who are illiterate, the scale will be administered by the research scholar. The postpartum mothers visiting the Immunization Clinic will undertake the pre-test, screened for PPD and later will be made to watch the video.

After watching the video, a feedback form will be provided to the mothers regarding the effectiveness of the video. The post-test will be conducted at an interval of 4–8 weeks of time after pre-test [35] to ensure knowledge retention and also the compliance for posttest, as the mothers will be visiting immunization clinics at 6, 10 and 14 weeks for vaccination of the child as part of the Universal Immunization Program [36]. Lastly the mothers will be re-screened for PPD using PHQ 9. In case a mother scores high on PHQ 9, indicating depression them they will be seen by the trained medical officer (As a part of District Mental Health Program (DMHP), all the PHCs runs 'Manochaitanya Clinic' on Tuesdays.) and later mothers will be referred to manochiatanya clinic for further evaluation and treatment by mental health professionals.

## Study outcome

### Primary outcome.

- This video intervention is intended to improve the awareness and reduce stigma on postpartum depression among mothers attending immunization clinics. The same will be assessed using the PoDLiS scale before and after the intervention.

- The developed educational video will be useful to help identify mothers with PPD which can then lead to early initiation of treatment and improved health outcomes for both mothers and their children. This will be done using the PHQ 9 before and 4–8 weeks after intervention.

### Secondary outcome.

- The developed video intervention can be utilised in future as an educational tool to improve perinatal mental health literacy and reduce stigma among expectant mothers, postpartum mothers, family members and community.

- The video can also be used for capacity building and training of health professionals and healthcare workers working in the area of perinatal mental health.

**Data management plan.** The collected data will be kept anonymous using unique IDs, data will be entered in a dedicated laptop which will be password protected and only the research team will be provided the access.

## Statistical analysis

The salient findings from literature search will be summarized and findings will be included for the script. The outcomes of FGDs will be qualitatively analysed. Inductive and thematic theme identification will be done using a descriptive approach. The analysis will be done manually and the themes will be generated based on the codes from the discussion. The findings will be utilized for finalising the video script. The effectiveness of the video intervention will be quantitatively analysed using paired t test. The results will be presented as mean, SD at each time point and 95% CI of the difference between pre and post means. Repeated measures of ANOVA will be used to analyse the change in Postpartum Depression Literacy Scale scores with respect to socio-demographic variables. The prevalence of PPD among mothers will be analysed using PHQ 9.

**Ethical considerations.** Ethical committee clearance is obtained from Institute Ethics Committee of NIMHANS and Ramaiah Medical College Hospital, Bangalore.

**The status and timeline of the study.** The timeline for the entire study is 27 months. The timeline allotted for recruitment and procurement is 3 months. The development of the video script, video shooting & editing will be completed in 9 months. The data collection will be done over 6 months and 9 months for data analysis & report preparation.

## Discussion

This is a Quasi-experimental single group study to evaluate the effectiveness of video intervention in improving awareness and reducing stigma on Postpartum Depression among postpartum mothers attending the immunisation clinics. The mental health literacy about perinatal mental health issues is poor and is also associated with stigma which affects the help seeking among mothers [9]. Postpartum depression is poorly understood in India and people fail to recognise it as a health problem and seek help from alternate sources like religious places [12]. There is a lack of availability of educational interventions and information resources on PPD in regional languages relevant to Indian cultural context [31]. The public's perception of perinatal mental health issues is one aspect that must be taken into account in order to implement interventions that have the envisioned outcomes. Several studies suggest that the women have poor knowledge about PPD [32] which could potentially discourage them from seeking professional help. It may be possible to avoid stigma related behaviours by using innovative approaches, as it has been demonstrated that internet-based methods which include knowledge and cognitive behavioural techniques can influence attitudes regarding PPD and levels of stigma [22, 33].

If this video intervention is found to be effective in improving awareness and reducing stigma on PPD among postpartum mothers, then it can be utilised for early identification and management of PPD among mothers. This could also be useful in achieving better maternal mental health and child health outcomes for the country. The video intervention is mainly aimed at improving awareness, reducing stigma and prevention of PPD through good family support, adopting healthy lifestyles, access to information, self-care and improved help seeking.

### Limitations of the study

The study includes a single group (intervention) and there is no control group to compare the effectiveness of video intervention. The outcome includes only the improvement in knowledge and reduction of stigma on PPD among mothers and does not include assessment of any health-related outcomes such as improved quality of life and prevention of relapse. The mothers are recruited from the immunization clinics at the urban and rural health centres attached to the medical college and not be the actual representation of the community affecting the generalisability of the study findings.

### Dissemination of findings

The video intervention can be used as a capacity building model for healthcare workers in maternity homes, immunization clinics, health & wellness centres and field workers like Accredited Social Health Activist (ASHA) and Auxiliary Nurse Mid-wife (ANM). The video intervention will be developed in different regional languages for wider reach. The video can be showcased through mass media, social media to create awareness and aid in early detection and management of Postpartum Depression. The findings will also be shared in the form of scientific reports and publications.

## Supporting information

**S1 Checklist. SPIRIT 2013 checklist: Recommended items to address in an Interventional trial protocol and related documents\*.**
(DOCX)

## Acknowledgments

The authors would like to thank Ms. Deepika Saini, Former fellowship student, Department of Mental Health Education, NIMHANS for her contributions in the preparation of the grant proposal.

## Author Contributions

**Conceptualization:** Latha K., Sundarnag Ganjekar.

**Data curation:** Latha K., Mariamma Philip, Swathi Acharya.

**Formal analysis:** Latha K., Sundarnag Ganjekar, Mariamma Philip.

**Funding acquisition:** Latha K.

**Investigation:** Latha K., Suman G., Dinesh Rajaram, Swathi Acharya.

**Methodology:** Latha K., Sundarnag Ganjekar, Meena K. S., Virupaksha H. S., Mariamma Philip.

**Project administration:** Latha K., Swathi Acharya.

**Resources:** Latha K., Sundarnag Ganjekar, Meena K. S., Virupaksha H. S., Mariamma Philip, Suman G., Dinesh Rajaram.

**Supervision:** Latha K., Sundarnag Ganjekar, Meena K. S.

**Validation:** Latha K., Sundarnag Ganjekar, Meena K. S., Virupaksha H. S., Mariamma Philip, Suman G., Dinesh Rajaram, Kimneihat Vaiphei.

**Writing – original draft:** Latha K., Swathi Acharya.

**Writing – review & editing:** Latha K., Sundarnag Ganjekar, Meena K. S., Virupaksha H. S., Mariamma Philip, Suman G., Dinesh Rajaram, Kimneihat Vaiphei, Somshekhar A. R.

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
