## [Decision Letter · Decision Letter 0]

5 Jan 2024

PONE-D-23-15242Study on Awareness and Management based Health Action using Video Intervention (SAMBHAV) for Postpartum Depression among mothers attending immunisation clinic in a tertiary medical college hospitalPLOS ONE

Dear Dr. Latha

Thank you for submitting your manuscript to PLOS ONE. After careful consideration, we feel that it has merit but does not fully meet PLOS ONE’s publication criteria as it currently stands. Therefore, we invite you to submit a revised version of the manuscript that addresses the points raised during the review process.

We look forward to receiving your revised manuscript.

Kind regards,

Pracheth Raghuveer, MD, DNB

Academic Editor

PLOS ONE

Journal Requirements:

2. Please ensure that you include a title page within your main document. You should list all authors and all affiliations as per our author instructions and clearly indicate the corresponding author.

3. Please ensure that you refer to Figure 1 in your text as, if accepted, production will need this reference to link the reader to the figure.

Reviewers' comments:

Reviewer's Responses to Questions

**Comments to the Author**

1. Does the manuscript provide a valid rationale for the proposed study, with clearly identified and justified research questions?

Reviewer #1: Yes

Reviewer #2: Yes

Reviewer #3: Partly

Reviewer #4: Yes

2. Is the protocol technically sound and planned in a manner that will lead to a meaningful outcome and allow testing the stated hypotheses?

Reviewer #1: Yes

Reviewer #2: Yes

Reviewer #3: No

Reviewer #4: Yes

3. Is the methodology feasible and described in sufficient detail to allow the work to be replicable?

Reviewer #1: Yes

Reviewer #2: Yes

Reviewer #3: No

Reviewer #4: Yes

4. Have the authors described where all data underlying the findings will be made available when the study is complete?

Reviewer #1: Yes

Reviewer #2: Yes

Reviewer #3: No

Reviewer #4: Yes

5. Is the manuscript presented in an intelligible fashion and written in standard English?

Reviewer #1: Yes

Reviewer #2: Yes

Reviewer #3: Yes

Reviewer #4: Yes

6. Review Comments to the Author

You may also provide optional suggestions and comments to authors that they might find helpful in planning their study.

Reviewer #1: The manuscript covers a important research topic. A interesting recent published article may be of interested in utilized for comparative analysis:

Chaudhary K, Nepal J, Shrestha K, Karmacharya M, Khadka D, Shrestha A, Shakya PR, Rawal S, Shrestha A. Effect of a social media-based health education program on postnatal care (PNC) knowledge among pregnant women using smartphones in Dhulikhel hospital: A randomized controlled trial. Plos one. 2023 Jan 20;18(1):e0280622.

A minor suggestion:

Discussion: Please kindly highlight more clearly the limitations of the study with a few justifications particularly the generalizability of the findings.

Reviewer #2: This is an important study with high practical implication. This proposal seemed to be well structured and well planned.

Reviewer #3: This protocol addresses a significant public health concern: poor perinatal mental health exacts a severe and substantial toll borne by women, their families, and the healthcare system. The authors’ intention to address this issue by raising awareness and lessening the stigma of postpartum depression is laudable.

Areas for revision and clarification:

Pg. 2 “the percentage of women reporting difficulty in mother child bonding and harm to the baby varied between 5- 77.1% [9,10]” – Suggest revision for clarity; this could be easily misinterpreted that up to 77% of women harm their baby.

Pg. 3 – regarding the birth of a female baby as a risk factor for PPD, please clarify whether this is globally or within a particular geographic/cultural context.

Pg. 3 – “Postpartum Depression affects the bonding between the mother and her child and also impairs growth and development of the baby. This might predispose her to chronic or recurring depression.” Citation is needed here. Please clarify the link between diminished bonding and its impacts on infant developmental trajectories and how these factors predispose mothers to chronic or recurring depression. Perhaps the phrase “This might predispose her to chronic or recurring depression” is intended to appear after the discussion on barriers to treatment.

Pg. 5 While the sample size for the quasi-experimental component of the study (pre and post-testing educational video effectiveness) is justified via power calculations, no rationale is provided for the sample size of the focus groups or expert evaluation of the video scripts.

Please describe the psychometric properties of both the Postpartum Depression Literacy Scale and the Patient Health Questionnaire 9. Please include any details pertaining to whether these scales have been validated for use within Indian populations (e.g., are they culturally relevant to your target population).

In describing the literature search strategies for developing the script for the 2 videos that form the basis of this intervention, the authors should expand on their stated search parameters. For instance, will the search include grey literature as well as peer-reviewed; any limits with respect to publication date? If the authors are looking for literature that encompasses the Indian context, why is the search limited to English-language only? What search strategies will the authors use to access “blogs”? Furthermore, the authors should consider whether the quality of the included articles will be assessed/appraised in some way.

Pg. 7 The following passage appears twice on this page: “Focus group discussion will be conducted among the mother’s attending immunization clinics and healthcare professionals including both doctors and non-doctors including Psychiatric Social Work, Clinical Psychology, Nursing, Obstetrics, Paediatrics and health workers. This will help in-depth understanding about postpartum depression at the grass root level. This will also help in finalising the content for the video script which is a crucial step in the study.” – It is unclear whether this refers to two distinct study phases in the study or is repeated in error.

In the focus group discussions that will support developing the video script, the authors should provide details about the number participants in each focus group, the length of time of the focus, and how many focus group meetings they anticipate conducting. What participants will comprise each focus groups – e.g., mothers and health professionals interviewed separately or together? What is the nature or the questions that will guide the focus group discussions & what setting will these meetings occur?

Pg. 8: Details regarding the evaluation of the scripts are limited. How will face validity be determined? What is the nature of the data being collected (surveys, qualitative interviews)? How will this data then be analyzed? The authors note that experts, mothers, & caregivers will be evaluating the scripts but do not provide details about how many informants will participate. This is also the first mention of “caregivers attending perinatal clinics.” If caregivers are to be included in the study population, then details should be provided in the appropriate section.

What is the rationale for the 2-month follow-up in administering the postpartum depression literacy scale? Is it expected that mothers will retain the educational information about PPD provided in the videos for that length of time after only one viewing of the video?

Furthermore, as the other outcome of this video series is for early identification and management of PPD, it also isn’t clear how this will be accomplished after one viewing.

Pg. 9: under statistical analysis “Based on the literature search, important components will be identified to create the script for the video.” – it is unclear what this means. Also, there is no mention of how the PHQ – 9 will be analysed.

For the focus group discussion data, the type of thematic analysis is not specified (e.g., Braun & Clark) and the details of the coding strategy are not discussed.

There do not appear to be any details about recruitment strategies or study setting beyond “immunization clinics.

Reviewer #4: Thanks for allowing me to review this excellent study protocol. It's quite thorough, but I do advise a few minor modifications:

1.PLOS ONE says they require a plan for data to be available to readers. You mention your plan for data in the abstract portion as "N/A" since "no results are reported." I do see that in the body of the manuscript (on p1 just before the abstract), you do address data availability differently: "After the study is completed, all relevant data will be made available." Should reconcile these two.

2. In the abstract Discussion section line 2, authors mention that the video "can be used to help mothers identify and treat PPD early on improving both maternal and child health outcomes." I appreciate that the video will address knowledge and hence identification of PPD (for mothers and for healthcare staff/workers), but it would not offer treatment. Should therefore remove "and treat" from above. Furthermore, as authors are not measuring health outcomes in this project, I advise a less bold claim in that same sentence regarding outcomes, something like: "...it can be used to help identify mothers' PPD, which can then lead to treatment and improved health outcomes for both mothers and their children."

3. Methods and Analysis section, under Exclusion criteria (p5), I presume you will allow mothers to opt out from completing the surveys (PHQ9 and PoDLiS) and from watching the video? You could also add that speaking/reading languages other than English and Kannada are exclusions.

4. Methods and analysis section, sample Size (p5): It appears that you are looking to recruit 130 mothers over a period of 27 months? That's just about 5 patients a month. Help me understand (as the reader) why it should be so hard to recruit these mothers? Is it the shortage of postpartum mothers within 6 months of birth attending the immunization clinic? Is it the lack of mothers speaking or reading English or Kannada? Do you anticipate that most mothers will opt out due to the stigma associated with PPD?

5. There are several typos I noted, including "et al." which is used multiple times through the research manuscript and should have a period following it. Another example is "immunization" (eg., top of p11) vs. "immunisation." One is a more Indian/British spelling. I believe either is OK, but there should be consistency across the manuscript. Have some colleagues help proofread for such errors slipping through.

6. Development of Script (pp 7-8): There is no mention of consideration of length of the script and resultant video. Presumably the length would impact the compliance of women with watching it. Consider some mention of balancing the important content with the resultant length.

7. Administering the questionnaire to mothers (p8): Please elaborate whether the 2-month delayed post-test is designed to assess retention of the information regarding PPD or whether it is intended to pick up new development of PPD 2-months later?

8. Consistency across the paper: Sometimes the video intervention is mentioned as a single video (eg., Study Outcome section on pp 8-9), and sometimes as two videos (eg., top of p7 in the Data Collection section).

9. Unclear acronym: "FGDs" in Quantitative Analysis section p9. Maybe I missed it, but I'm unclear what this acronym refers to. Perhaps the focus group discussions (per google search, but I couldn't guess without internet help)?

10. Limitations section on p10: I believe this should be written in paragraph format, consistent with the rest of the research proposal manuscript. I agree that the study is limited by choosing not to measure any health outcomes. Consider adding one or more survey questions to the 2-month follow up... If a mother does score highly on the PHQ9 indicating depression, does she intend to seek mental health support services? If so, from where? If not, why?

7. PLOS authors have the option to publish the peer review history of their article (what does this mean?). If published, this will include your full peer review and any attached files.

Reviewer #1: No

Reviewer #2: **Yes: **Panchanan Acharjee

Reviewer #3: No

Reviewer #4: No

---

## [Author Response · Author response to Decision Letter 0]

14 Feb 2024

We have gone through the comments shared by the reviewers and made changes accordingly. The same has been elaborated in the Response to Reviewers File.

Thank you for the valuable insights provided. We look forward to hear from you.

---

## [Decision Letter · Decision Letter 1]

14 Mar 2024

Study on Awareness and Management based Health Action using Video Intervention (SAMBHAV) for Postpartum Depression among mothers attending immunisation clinic in a tertiary medical college hospital

PONE-D-23-15242R1

Dear Dr. Latha K,

We’re pleased to inform you that your manuscript has been judged scientifically suitable for publication and will be formally accepted for publication once it meets all outstanding technical requirements.

Kind regards,

Pracheth Raghuveer, MD, DNB

Academic Editor

PLOS ONE

Additional Editor Comments (optional):

Reviewers' comments:

Reviewer's Responses to Questions

**Comments to the Author**

1. Does the manuscript provide a valid rationale for the proposed study, with clearly identified and justified research questions?

Reviewer #1: Yes

2. Is the protocol technically sound and planned in a manner that will lead to a meaningful outcome and allow testing the stated hypotheses?

Reviewer #1: Yes

3. Is the methodology feasible and described in sufficient detail to allow the work to be replicable?

Reviewer #1: Yes

4. Have the authors described where all data underlying the findings will be made available when the study is complete?

Reviewer #1: Yes

5. Is the manuscript presented in an intelligible fashion and written in standard English?

Reviewer #1: Yes

6. Review Comments to the Author

You may also provide optional suggestions and comments to authors that they might find helpful in planning their study.

Reviewer #1: Thank you for addressing the points which were highlighted by reviewers previously. The manuscript reads well now.

7. PLOS authors have the option to publish the peer review history of their article (what does this mean?). If published, this will include your full peer review and any attached files.

Reviewer #1: No

---

## [Editor Report · Acceptance letter]

25 Mar 2024

PONE-D-23-15242R1 

PLOS ONE

Dear Dr. K, 

I'm pleased to inform you that your manuscript has been deemed suitable for publication in PLOS ONE. Congratulations! Your manuscript is now being handed over to our production team.

Kind regards, 

on behalf of

Dr. Pracheth Raghuveer 

Academic Editor

PLOS ONE